# Uncovering Universal Features: How Adversarial Training Improves Adversarial Transferability

**Jacob M. Springer** [1]  **Melanie Mitchell** [2]  **Garrett T. Kenyon** [1]

## Abstract

Adversarial examples for neural networks are known to be transferable: examples optimized to be misclassified by a "source" network are often misclassified by other "destination" networks. Here, we show that training the source network to be "slightly robust"—that is, robust to small-magnitude adversarial examples—substantially improves the transferability of targeted attacks, even between architectures as different as convolutional neural networks and transformers. In fact, we show that these adversarial examples can transfer representation (penultimate) layer features substantially better than adversarial examples generated with non-robust networks. We argue that this result supports a non-intuitive hypothesis: slightly robust networks exhibit *universal* features—ones that tend to overlap with the features learned by all other networks trained on the same dataset. This suggests that the features of a single slightly-robust neural network may be useful to derive insight about the features of *every* non-robust neural network trained on the same distribution.

## 1. Introduction

Neural-network image classifiers are well-known to be susceptible to adversarial examples—images that are perturbed in a way that is largely imperceptible to humans but that cause the neural network to make misclassifications. Many explanations have been offered for this susceptibility as well as for the transferability of adversarial examples across network architectures and even training sets; however, the ML community's understanding of these phenomena remains incomplete.

Here, we find that the features of slightly-robust convolu-

tional neural networks substantially overlap with the features of *every* tested non-robust network, including transformer networks (ViT and CLIP (Dosovitskiy et al., 2020; Radford et al., 2021)), which differ substantially in architecture, and in the case of CLIP, in the training objective. As a practical matter, this means that targeted adversarial examples—which aim to fool a network into misclassifying the example as a chosen class—constructed with slightly-robust "source" networks are highly transferable to non-robust "destination" networks, regardless of architecture. In other words, the features of slightly-robust networks are *universal* with respect to the features of non-robust networks (Olah et al., 2020). This notion of universality can explain why slightly-robust networks give rise to more transferable attacks and have better weight initialization for downstream transfer-learning tasks (Liang et al., 2020; Salman et al., 2020; Terzi et al., 2020; Utrera et al., 2020). In addition, the phenomenon suggests that the features of a single slightly-robust network may be useful to derive insight about the features of *every* non-robust network.

## 2. The Exceptional Transferability of Adversarial Examples Generated with Slightly-Robust Classifiers

In this section, we describe the methodology and results of our experiments on the transferability of adversarial examples when generated with a adversarially-trained source networks. Adversarial training involves adversarially perturbing the input at the training step (Madry et al., 2017). The training is parameterized by a value $\varepsilon$, which specifies the maximum $\ell_2$ norm of the adversarial perturbation. We call this parameter the *robustness parameter* of the network and say that a network is $\varepsilon$-robust when it has a robustness parameter of $\varepsilon$. Our goal is to quantify how the robustness parameter of the source network affects the transferability of adversarial examples. We evaluate the targeted transferability of adversarial examples to destination convolutional networks Xception (Chollet, 2017), VGG (Simonyan & Zisserman, 2014), ResNet (He et al., 2016a;b), Inception (Szegedy et al., 2016), MobileNet (Howard et al., 2017), DenseNet (Huang et al., 2017), NASNetLarge (Zoph et al., 2018), and EfficientNet (Tan & Le, 2019), as well as two

[1]Los Alamos National Laboratory, Los Alamos, NM [2]Santa Fe Institute, Santa Fe, NM. Correspondence to: Jacob M. Springer <jacmspringer@gmail.com>.

*Accepted by the ICML 2021 workshop on A Blessing in Disguise: The Prospects and Perils of Adversarial Machine Learning.* Copyright 2021 by the author(s).

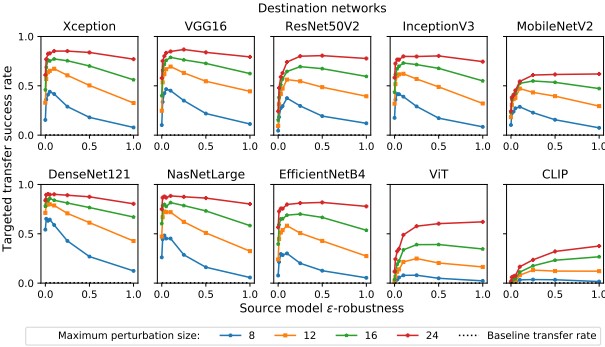

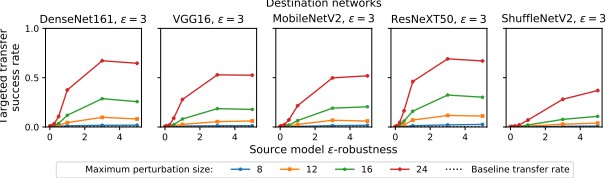

*Figure 1.* Targeted transfer attack success rate against ImageNet classifiers using $\varepsilon$-robust ResNet50 source models. The targeted transferable adversarial examples are limited by a maximum perturbation size $\|\delta\|_\infty \leq {}^{16}/_{256}$. Higher is a more successful attack. The baseline transfer rate refers the rate at which the original (unperturbed) images are classified as the target class. (Best viewed in color.)

transformer-based networks, ViT (Dosovitskiy et al., 2020) and CLIP (Radford et al., 2021).

**Generating Adversarial Examples.** We generate targeted adversarial examples using 1000 randomly selected images from the ImageNet validation dataset and 1000 randomly selected target classes. We generate adversarial examples for each image-target pair, for each source network (of different $\varepsilon$-robustness). We run the TMDI-FGSM algorithm for 300 iterations to generate each perturbation (Zhao et al., 2020). We detail the exact process in the appendix.

**Transferability to Convolutional Networks.** In Figure 1, we plot the transfer success rate for the targeted adversarial examples. We include the baseline performance of the attacks that would be measured if the images were unperturbed. We find that across every convolutional destination network, a source network with a small robustness parameter ($\varepsilon \approx 0.1$) improves the transfer success rate when compared to the non-robust ($\varepsilon = 0$) and more-robust ($\varepsilon > 0.5$) source networks (Figure 1).

**Transferability to Transformer-Based Classifiers.** Few studies have addressed the robustness of transformer-based classifiers to transfer attacks (Shao et al., 2021). To our knowledge, our paper is the first to address *targeted* transfer attacks against transformer architectures. We find a striking result: targeted adversarial examples constructed with non-robust convolutional source networks are almost entirely non-transferable to transformer networks (Figure 1). This suggests that the features learned by transformer-based models and non-robust convolutional models are largely different. However, we find that using a *slightly-robust* ResNet50 classifier as a source network dramatically im-

*Figure 2.* Targeted transfer attack success rate against destination ($\varepsilon = 3$)-robust ImageNet classifiers using $\varepsilon$-robust ResNet50 source networks. The targeted transferable adversarial examples are limited by a maximum perturbation size $\|\delta\|_\infty \leq {}^{16}/_{255}$. Higher is a more successful attack. Baseline refers the rate at which unperturbed images are classified as the target class. (Best viewed in color.)

proves the transferability of targeted adversarial examples to transformer-based classifiers. The optimal robustness parameter to maximize transferability is larger when the destination network is a transformer ($\varepsilon \approx 1$ than when it is convolutional ($\varepsilon \approx 0.1$).

**Attacking Adversarially-Trained Models.** Adversarial training has been shown to improve robustness to transfer attacks (Madry et al., 2017). We evaluate the transferability of adversarial examples on adversarially trained destination networks. Even though the adversarial perturbations which we use to attack each adversarially-trained network are larger than the magnitude for which the destination networks are trained to be robust, the adversarial examples generated using non-robust ($\varepsilon = 0$) networks do not transfer to the robust networks. However, as the robustness of the source network increases, the attack success rate increases substantially. Similarly to Figure 1, Figure 2 includes the baseline performance of the attacks that would be measured if the images were unperturbed.

## 3. Universality of Slightly-Robust Features

The previous section established that slightly-robust networks can be used as source networks to construct highly transferable adversarial examples. In this section, we examine the reason why adversarial examples transfer successfully.

It has been previously shown that robust and non-robust features overlap to some degree (Springer et al., 2021). Intuitively, this overlap is speculated to arise, for example, when a non-robust feature identifies a local pattern, such as the texture of a dog ear, and when a robust feature identifies a more global pattern, such as an entire dog. However, there may be many equally-predictive local features. For example two local features, one which responds to a dog nose and one a dog ear, may be equally predictive. Both would overlap with a more global feature that responds to an entire

dog, but not with each other. Since there may be many non-overlapping features that are equally predictive, a classifier may only learn a subset of them. A classifier which learns features that overlap substantially with the features of *every* other standard classifier of a certain type (e.g., non-robust and trained on the same distribution) is said to have features that are *universal* with respect to this type (Olah et al., 2020). We speculate that non-robust and robust neural networks learn local and global features, respectively, for example, due to the association between non-robust features and texture bias, and robust features and shape bias (Geirhos et al., 2018). Thus, we expect that the features of robust networks are *universal* with respect to non-robust networks trained on the same distribution. If true, this means not only that robust networks would transfer the classification, but also the features (i.e., representation layer activations).

### 3.1. Testing Universality of Slightly-Robust Features

We aim to address the question, *to what extent do adversarial examples transfer representation-layer features transfer across networks—that is, evoke analogous patterns of activation?* In particular, we show that adversarial examples constructed to attack slightly-robust source networks transfer features substantially more effectively to every tested destination network than non-robust networks. This result supports our hypothesis that slightly-robust networks rely on universal features.

Our experiments rely on two types of adversarial examples: (1) *class-targeted*, which are optimized to produce a specific incorrect class in a source network, and (2) *representation-targeted*, which are optimized to produce a similar pattern of activation in the representation layer as a specific input of a different class. We refer to the degree to which adversarial examples of a source model can analogously affect the features that are computed by the representation layer of the destination model as the *representation transferability* from source to destination. By contrast, *class transferability* (what is commonly referred to as just *transferability*) refers to the degree to which adversarial examples can analogously affect the output of the destination network.

**Measuring Representation Transferability and Universality.** To measure representation transferability, we generate representation-targeted adversarial examples by choosing two examples from a dataset, $y_0$ and $x$, with different classes, and applying the TMDI-FGSM algorithm (Zhao et al., 2020) to minimize the $\ell_2$ distance between the representation-layer activations in the source network evoked by the adversarial example, $y = y_0 + \delta$, and the target input $x$. We limit the $\ell_\infty$ norm of $\delta$ to the standard $16/255$. Both $x$ and $y$ share similar representation-layer activations in the source network. If $x$ and $y$ also produce similar representation-layer activations in the destination network,

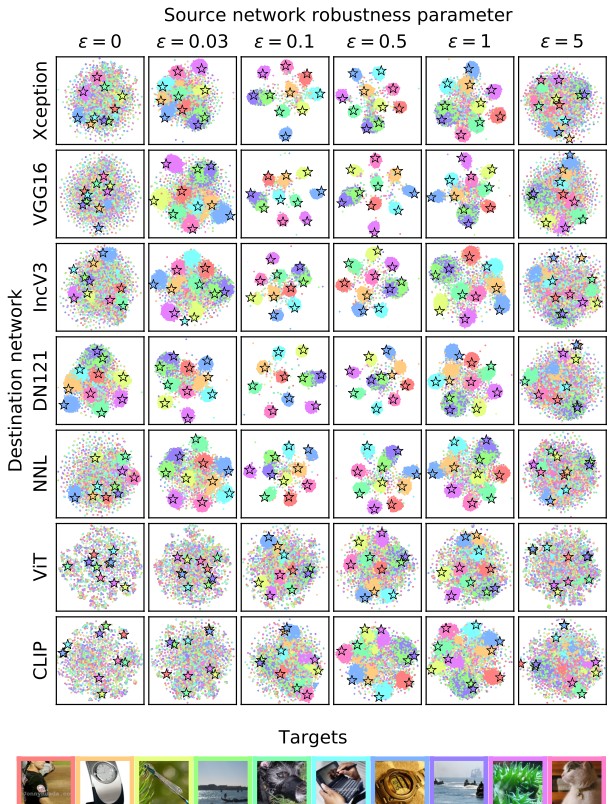

*Figure 3.* t-SNE plots of destination-network representations of representation-targeted adversarial examples generated by using whitebox ResNet50 models of specified $\varepsilon$-robustness. (Best viewed in color and magnified.)

then the representation transferability of the source network to the destination network is high. We select 990 initial images ($y_0$) and 10 targets ($x$), and construct representation-targeted adversarial examples for each pair, for a total of 9900. By measuring the similarity of the representations for each pair $(x, y)$ in the destination network, we can measure the representation transferability of the source network.

### 3.2. Results

We plot the $t$-distributed stochastic neighbor embedding (t-SNE) of the representation vectors of each representation-targeted adversarial example $y$ in the destination network (Figure 3). Each color corresponds to one of the 10 target images ($x$). The 10 stars in each plot correspond to the t-SNE embedding of the destination-network representation of each target $x$. When representation transferability is high, points associated with representation-targeted adversarial examples group tightly next to the star associated with their target (in Figure 3, points group by color). We observe that slightly-robust source networks exhibit a high degree of representation transferability across *every* tested model,

*Table 1.* Cosine similarity between representations (in the destination network) of representation-targeted adversarial examples $y$ and the corresponding target image $x$, as a function of robustness parameter $\varepsilon$ of the source network. Each value is an average over the 9900 $(x, y)$ pairs of representation vectors.

| | Source network robustness parameter ($\varepsilon$) | | | | | | | | | |
|---|---|---|---|---|---|---|---|---|---|---|
| Dest | 0 | 0.01 | 0.03 | 0.05 | 0.1 | 0.25 | 0.5 | 1 | 3 | 5 |
| Xcept | 0.46 | 0.50 | 0.53 | 0.56 | **0.59** | 0.58 | 0.57 | 0.54 | 0.44 | 0.40 |
| VGG16 | 0.33 | 0.40 | 0.41 | 0.49 | **0.52** | 0.52 | 0.52 | 0.48 | 0.38 | 0.33 |
| RN50 | 0.28 | 0.34 | 0.37 | 0.43 | 0.49 | 0.49 | **0.51** | 0.48 | 0.38 | 0.32 |
| IncV3 | 0.57 | 0.61 | 0.62 | 0.64 | **0.67** | 0.66 | 0.65 | 0.63 | 0.57 | 0.53 |
| MNv2 | 0.43 | 0.45 | 0.46 | 0.49 | **0.52** | 0.51 | 0.51 | 0.50 | 0.46 | 0.43 |
| DN121 | 0.67 | 0.69 | 0.69 | 0.71 | **0.73** | 0.71 | 0.71 | 0.68 | 0.62 | 0.58 |
| NNL | 0.36 | 0.42 | 0.45 | 0.49 | **0.54** | 0.51 | 0.48 | 0.44 | 0.32 | 0.27 |
| ENB4 | 0.09 | 0.11 | 0.14 | 0.14 | **0.24** | 0.22 | 0.23 | 0.20 | 0.11 | 0.07 |
| ViT | 0.07 | 0.09 | 0.11 | 0.13 | 0.20 | **0.21** | **0.21** | 0.20 | 0.12 | 0.09 |
| CLIP | 0.53 | 0.54 | 0.55 | 0.56 | 0.59 | 0.51 | **0.61** | **0.61** | 0.58 | 0.57 |

in comparison with non-robust and more-robust source networks. This suggests that slightly-robust source networks rely on universal features—features that are useful to *every* tested non-robust destination model, regardless of architecture, and, strikingly, useful even to CLIP, whose training objective seeks to associate images to textual descriptions rather than to a discrete set of labels. Our results provide empirical evidence that universal features exist and that networks can use adversarial training with a small robustness parameter to learn these universal features.

As a second similarity metric, we report the mean cosine similarity of the representations (in destination networks) between $(x, y)$ pairs, averaged over all 9900 such pairs for each source network (Table 1). This similarity peaks for destination CNNs when the source network has $\varepsilon = 0.1$, except for ResNet50v2, which peaks at $\varepsilon = 0.5$. Destination network ViT peaks at $\varepsilon \in \{0.25, 0.5\}$ and CLIP when $\varepsilon \in 0.5, 1$. These peaks are consistent with our t-SNE results (Figure 3), and are included primarily for the purpose of quantitatively confirming the visual results of our t-SNE plots.

## 4. Related Work

The vulnerable features learned by neural networks have been studied both empirically (Aubry & Russell, 2015; Dosovitskiy & Brox, 2016; Geirhos et al., 2020; Ilyas et al., 2019; Jo & Bengio, 2017; Mahendran & Vedaldi, 2015; McCoy et al., 2019; Olah et al., 2017; Simonyan et al., 2013; Wang et al., 2020; Wei, 2020; Zhang & Zhu, 2018) and theoretically (Allen-Zhu & Li, 2020; Arpit et al., 2017; De Palma et al., 2019; Hermann et al., 2020; Nakkiran et al., 2019; Shah et al., 2020; Valle-Pérez et al., 2018; Wu et al., 2017). There has been some research related to similarity of learned features across networks (Kornblith et al., 2019;

Li et al., 2015; Olah et al., 2020; Raghu et al., 2017). In addition, robust networks have been shown to have a number of valuable properties, including serving as a good starting point for transfer learning (Liang et al., 2020; Salman et al., 2020; Terzi et al., 2020; Utrera et al., 2020) and gradient interpretability (Engstrom et al., 2019b). In addition, there has been substantial work on transferable adversarial example (Carlini & Wagner, 2017; Dong et al., 2018; 2019; Goodfellow et al., 2014; Guo et al., 2020; Huang et al., 2019; Inkawhich et al., 2019; 2020a;b; Kurakin et al., 2016; Li et al., 2020; Liu et al., 2016; Papernot et al., 2016; Rozsa et al., 2017; Sharma et al., 2019; Song et al., 2018; Tramèr et al., 2018; Wu et al., 2020; Xie et al., 2019; Zhao et al., 2017; 2020; Zhou et al., 2018).

## 5. Conclusion

In this paper, we have shown that slightly-robust models learn features that overlap substantially with *every* tested non-robust model; such features have been termed *universal* (Olah et al., 2020). This overlap leads to representation transferability, which can be used to construct targeted transferable adversarial examples that transfer to across substantially different architectures (ViT) and even training objectives (CLIP). Since most previous transferable adversarial generation techniques rely on optimizing adversarial examples over a non-robust source network, our technique can be combined with virtually any previously existing optimization technique by replacing the non-robust source network with a slightly-robust network.

More generally, our paper reveals a phenomenon that is significant for the broader field of deep learning: we find that different non-robust networks, even when trained with similar convolutional architectures, do not necessarily have many features that substantially overlap. This can have important implications for the reliability of neural networks; when different networks rely on different features, they are susceptible to different types of errors. In addition, we present an argument that, for a given task, there are features that are useful to every tested neural network, and that these features can be learned with small-$\varepsilon$ adversarial training, even when the source network architecture and learning objective are dissimilar to those of the destination network. Thus, by studying the features of a single slightly-robust network, we can empirically discover properties that will be applicable across all non-robust networks. We speculate that this phenomenon can explain why slightly-robust networks are successful at transfer-learning tasks (Liang et al., 2020; Salman et al., 2020; Terzi et al., 2020; Utrera et al., 2020). With applications across the field of machine learning, the contributions of this paper provide an important stepping stone towards discovering a general understanding of the features learned by neural networks.

## Acknowledgements

Research presented in this article was supported by the Laboratory Directed Research and Development program of Los Alamos National Laboratory under project number 20210043DR.

Melanie Mitchell's contributions were supported by the National Science Foundation under Grant No. 2020103. Any opinions, findings, and conclusions or recommendations expressed in this material are those of the author and do not necessarily reflect the views of the National Science Foundation.

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

# A. Detailed Methods

**Adversarial Examples.** In this paper we are primarily concerned with properties of source networks that facilitate transferability of adversarial examples. Let $f : \mathcal{X} \to \mathcal{Y}$ denote a "white-box" network (i.e., one whose architecture and weights are known to the adversary) and let $g : \mathcal{X} \to \mathcal{Y}$ denote a "black-box" network (weights and architecture are unknown to the adversary). Let $(x, y) \in \mathcal{X} \times \mathcal{Y}$ be an (unperturbed) input-label pair, where $\mathcal{X}$ is the input-space and $\mathcal{Y}$ is the label-space. Given a maximum perturbation size $\varepsilon$, our goal is to optimize over $f$ to construct a transferable adversarial example $x + \delta$ where $\|\delta\|_\infty \le \varepsilon$ and such that $g(x + \delta) \ne y$ for the *untargeted* case, and $g(x + \delta) = t$ for some target class $t \in \mathcal{Y}$ for the *targeted* case.

**Optimizers.** Prior research has identified a number of methods for optimizing adversarial examples given a white-box classifier $f$, many of which are based on the Iterative Fast Gradient Sign Method (I-FGSM) (Goodfellow et al., 2014; Kurakin et al., 2016), in which a perturbation $\delta_i$ is iteratively updated to maximize the loss of the network while obeying an $\ell_\infty$ norm constraint:

$$\delta_{i+1} = \delta_i + \alpha \cdot \text{sign} \, \nabla_{\delta_i} L(x + \delta_i),$$

where $L$ represents the adversarial loss function, $\alpha$ is a tunable step-size parameter, and $x + \delta_n$ is the final adversarial example after $n$ steps. At each step, $\delta_i$ is clipped such that $\|\delta\|_\infty \le \varepsilon$ and $x + \delta_i$ is a valid image.

We adopt the state-of-the-art method recently proposed by (Zhao et al., 2020), which combines three variants of I-FGSM and optimizes over many steps:

1. Diverse Input Iterative Fast Gradient Sign Method (DI$^2$-FGSM), which applies a random affine transformation to the input at each step prior to computing the gradient (Xie et al., 2019),
2. Translation-Invariant Iterative Fast Gradient Sign Method (TI-FGSM), which convolves the gradient with a Gaussian filter (Dong et al., 2019),
3. Momentum Iterative Fast Gradient Sign Method (MI-FGSM), in which a momentum term is added to the gradient (Dong et al., 2018).

Together called TMDI-FGSM, this optimization method can be described by the following process:

$$g_{\text{DI}}^{(i)} = \nabla_{\delta_i} L(T_i(x + \delta_i)) \qquad \text{(DI}^2\text{-FGSM)}$$

$$g_{\text{TDI}}^{(i)} = \mathcal{N} * g_{\text{DI}}^{(i)} \qquad \text{(TI-FGSM)}$$

$$g_{\text{TMDI}}^{(i)} = \mu \cdot g_{\text{TMDI}}^{(i-1)} + \frac{g_{\text{DI}}^{(i)}}{\|g_{\text{DI}}^{(i)}\|_1} \qquad \text{(MI-FGSM)}$$

$$\delta_{i+1} = \delta_i + \alpha \cdot \text{sign} \, g_{\text{TMDI}}^{(i)}$$

where $L$ again represents the adversarial loss function, $T_i$ represents a random affine transformation, $\mathcal{N}$ represents a Gaussian convolutional filter, $\mu$ is a tunable momentum parameter, and $\alpha$ is a tunable step-size parameter, and $x + \delta_n$ is the final adversarial example over $n$ steps.

For targeted adversarial examples, the loss function $L$ should be maximized when the target label is predicted with high confidence. For untargeted adversarial examples, this occurs when the predicted label differs from the true label, and when the true label is given a low confidence. A number of adversarial loss functions have been proposed, including standard cross-entropy loss (Szegedy et al., 2014), CW loss (Carlini & Wagner, 2017), and feature-disruption loss (Inkawhich et al., 2020a). We use the highly-effective logit loss, proposed by (Zhao et al., 2020), which is maximized for targeted examples when the logit score for a target class (i.e., the value of the output neuron associated with the target class prior to the softmax operation) is maximized. Similarly, the untargeted version aims to minimize the logit score associated with the true class.

For the DI$^2$ component of the optimization algorithm, we use a random resize and crop operation where each image is resized by a factor selected uniformly between $3/4$ and $4/3$, and then cropped to be $224 \times 224$ pixels randomly, with $0$-valued padding where appropriate. Then, a random horizontal flip is applied. This is equivalent to the PyTorch code:

```
transforms.Compose([
    transforms.RandomResizedCrop(size=[224, 224],
```

```
                        scale=(3/4, 4/3),
                        ratio=(1., 1.)),
    transforms.RandomHorizontalFlip()
])
```

For the TI component, we apply a Gaussian filter to the gradient at each step, with the filter size of $5 \times 5$, and the standard deviation of the filter 1.

For the MI component, we use a momentum of 0.9.

For generating representation-targeted adversarial examples, we exclude the TI step, as we found that the representation-targeted adversarial examples were less transferable when it was included.

**Constructing Robust Source Networks.** We construct robust source networks by performing adversarial training (Goodfellow et al., 2014; Madry et al., 2017). We use projected gradient descent in order to find model parameters $\theta^*$ that minimize the following expression:

$$\theta^* = \arg\min_{\theta} \mathbb{E}_{(x,y) \in \mathcal{D}} \big[ \max_{\|\delta\|_2 \leq \varepsilon} \mathcal{L}(\theta, x + \delta, y) \big],$$

where $L(\theta, x, y)$ represents the cross-entropy loss of a network with parameters $\theta$ evaluated on input $x$ with label $y$. We subject the adversarial examples constructed in the inner optimization procedure to an $\ell_2$ norm constraint. We will call this constraint $\varepsilon$ the *robustness parameter* of a classifier, as it represents the ($\ell_2$) magnitude of the adversarial examples with respect to which the classifier is trained to be robust. Due to the high computational cost of adversarial training, we rely on pre-trained robust ResNet50 models that have been pre-trained on ImageNet (Russakovsky et al., 2015). For our experiments, we test classifiers with robustness parameters $\varepsilon \in \{0, 0.01, 0.03, 0.05, 0.1, 0.25, 0.5, 1, 3, 5\}$.

**Model Details.** We use a number of models for our experiments. For all robust networks trained on ImageNet, we use the pre-trained weights that are available on the GitHub page associated with Salman et al. (2020). For all convolutional destination models, we use pre-trained weights that are included with Keras (Chollet et al., 2015). For the ViT model trained on ImageNet, we use pre-trained weights from Melas-Kyriazi (2020). For the CLIP model, we use the code and weights associated with (Radford et al., 2021).

We train robust CIFAR-10 models with the Robustness library (Engstrom et al., 2019a). We train for 100 epochs using a batch size of 128. We include data augmentation. We optimize using standard stochastic gradient descent with momentum, using a learning rate of 0.01 and a momentum parameter of 0.9, as well as a weight decay of 0.0001. For adversarial training, we generate each adversarial example with 7 steps, using a step-size of $0.3 \times \varepsilon$ for the given robustness parameter of $\varepsilon$. For the ViT model trained on CIFAR-10, we use pre-trained weights associated with Dosovitskiy et al. (2020) and finetune on CIFAR-10 for 10 epochs.

All convolutional destination CIFAR-10 models were finetuned for 20 epochs from the pre-trained ImageNet weights that are included with Keras (Chollet et al., 2015).

## B. Comparison to Previously Published Transfer Attack Methods

We directly compare the targeted transfer attack success rate to previous state-of-the-art black-box attacks and find that our method substantially outperforms the previous methods under similar constraints (Table 2). In particular, we evaluate our method's attack performace with three different loss functions: standard cross-entropy loss (Xent), Poincaré distance with a triplet loss term (Po+Trip) (Li et al., 2020), and logit loss. We include a comparison with the feature distribution attack (FDA) (Inkawhich et al., 2019), however, FDA requires that we train multiple supplemental models for each individual target class, which would require thousands of supplemental models to attack all thousand classes of ImageNet. Thus, we do not perform a direct comparison and instead report the targeted transfer attack success rate that is reported in the original FDA paper (Inkawhich et al., 2019).

*Table 2.* Direct comparison of targeted transfer attack success rate between our technique (slightly-robust source models, i.e., $\varepsilon > 0$) and previously proposed strong baseline attacks (non-robust source models, i.e., $\varepsilon = 0$). We compare three different loss functions: cross-entropy, Poincaré distance combined with triplet loss, and logit loss. In addition, we report the success rate of FDA from the original paper (see text for discussion). We limit the $\ell_\infty$ norm of the adversarial examples to the standard value of $16/255$.

| | | Xcept | VGG16 | RN50v2 | IncV3 | MNv2 | DN121 | NNL | ENB4 | ViT | CLIP |
|---|---|---|---|---|---|---|---|---|---|---|---|
| Xent | $\varepsilon = 0$ | 10.4 | 9.6 | 4.6 | 10.6 | 6.4 | 40.5 | 13.1 | 6.8 | 0.8 | 0.1 |
| Po+Trip | $\varepsilon = 0$ | 20.8 | 15.2 | 10.0 | 23.0 | 11.6 | 59.3 | 31.2 | 14.2 | 1.3 | 0.3 |
| Logit | $\varepsilon = 0$ | 45.9 | 40.0 | 15.3 | 43.6 | 22.9 | 77.9 | 60.3 | 39.6 | 3.9 | 0.4 |
| FDA* | $\varepsilon = 0$ | – | 43.5 | – | – | 22.9 | 57.9 | – | – | – | – |
| Xent | $\varepsilon = 0.1$ | 54.0 | 59.4 | 45.8 | 50.8 | 32.1 | 78.8 | 66.0 | 41.1 | 8.6 | 2.4 |
| Po+Trip | $\varepsilon = 0.1$ | 59.1 | 57.9 | 53.0 | 56.5 | 39.2 | 78.4 | 72.6 | 45.1 | 11.4 | 3.3 |
| Logit | $\varepsilon = 0.1$ | **77.2** | **78.8** | 64.5 | **73.1** | **52.5** | **84.0** | **81.6** | **68.9** | 33.4 | 11.2 |
| Xent | $\varepsilon = 1$ | 60.4 | 69.3 | **66.6** | 58.2 | 46.7 | 69.9 | 61.3 | 56.9 | 29.9 | 19.9 |
| Po+Trip | $\varepsilon = 1$ | 48.5 | 54.4 | 60.2 | 49.5 | 39.9 | 62.6 | 53.3 | 45.0 | 22.0 | 12.4 |
| Logit | $\varepsilon = 1$ | 56.1 | 62.4 | 59.5 | 55.0 | 47.2 | 67.0 | 58.3 | 53.6 | **36.0** | **26.7** |

# C. Extended ImageNet Data

In this section, we present extended data from the ImageNet.

**Untargeted Adversarial Examples.**    We use the 1000 transferable adversarial examples generated to transfer to ImageNet classifiers and plot the transfer success rate when we treat the adversarial examples as untargeted, i.e., we consider every adversarial example which is misclassified by the destination classifier as a success (Figure 4). In addition, we include analogous results for adversarially trained models (Figure 5).

**Representation-targeted Adversarial Examples.**    We include the extended data for the t-SNE figure presented in the main paper. We plot the destination-network representations of representation-targeted adversarial examples for every destination network and source network robustness parameter that we test (Figure 6).

**Additional Tested Source-Network Robustness Parameters.**    In the main paper, we exclude certain values of $\varepsilon$ in the figures that illustrate the transferability of adversarial examples for clarity, so that the results from slightly-robust networks could be more easily seen. We include the extended results for both targeted and untargeted adversarial examples (Figures 7 and 8). We observe a decrease in transfer performance as robustness increases past the optimal point. We speculate that this arises from the fact that as robustness increases, smaller features on which non-robust neural networks rely are gradually thrown away, thus reducing transfer performance.

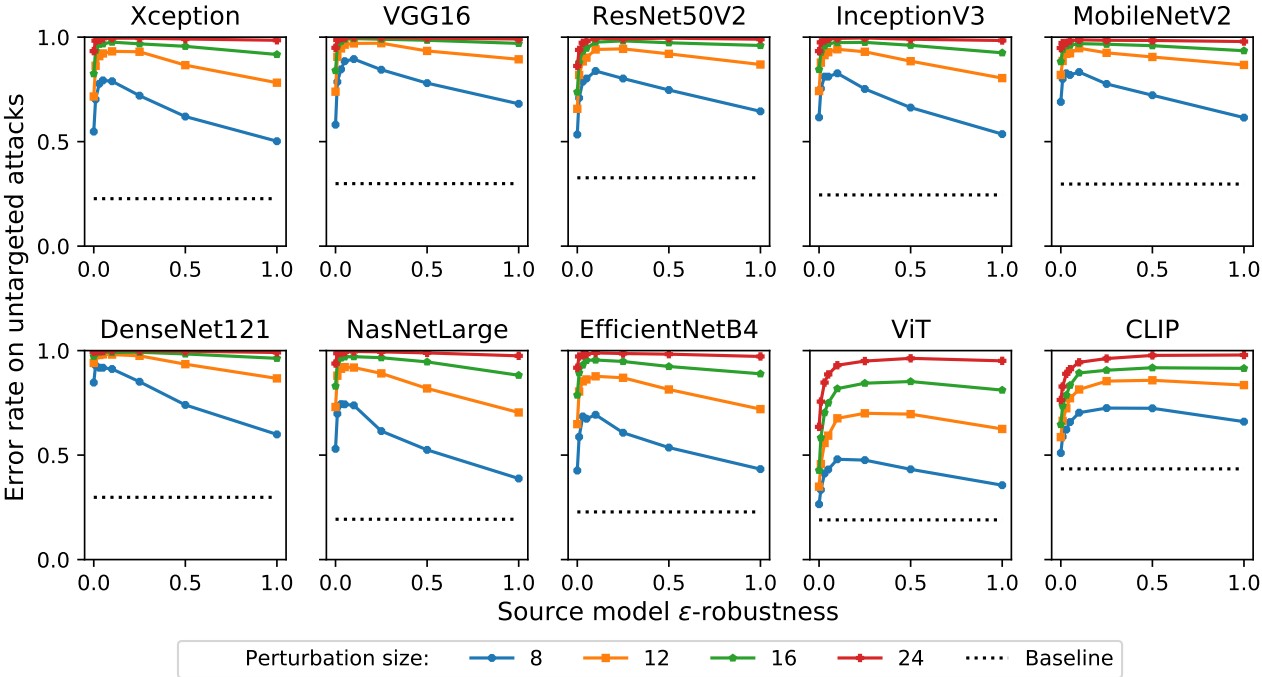

*Figure 4.* Error rate of destination networks (ImageNet classifiers) evaluated on untargeted transferable adversarial examples using $\varepsilon$-robust ResNet50 source models with perturbation size $\|\delta\|_\infty \leq {}^{16}/_{256}$. Higher is a more successful attack. Baseline refers to the misclassification rate of unperturbed images. (Best viewed in color.)

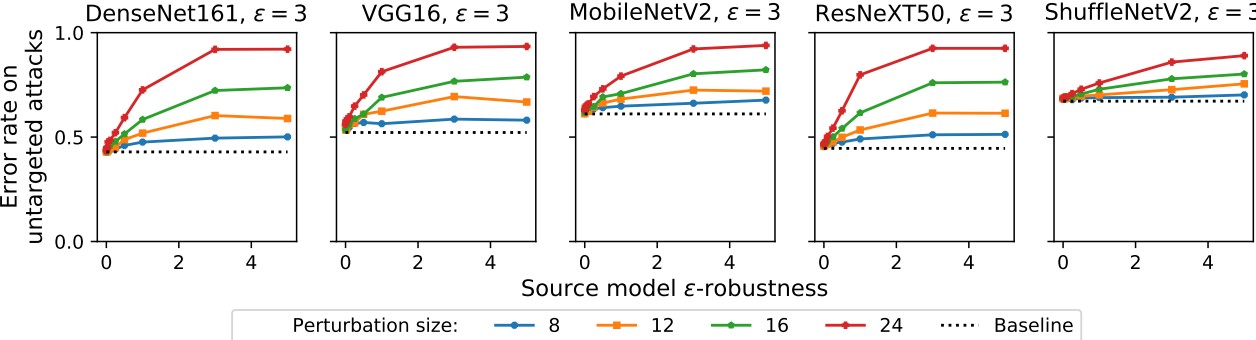

*Figure 5.* Error rate of destination ($\varepsilon = 3$)-robust ImageNet classifiers evaluated on untargeted adversarial examples using $\varepsilon$-robust ResNet50 source networks with perturbation size $\|\delta\|_\infty \leq {}^{16}/_{256}$. Higher is a more successful attack. Baseline refers to the misclassification rate of unperturbed images. (Best viewed in color.)

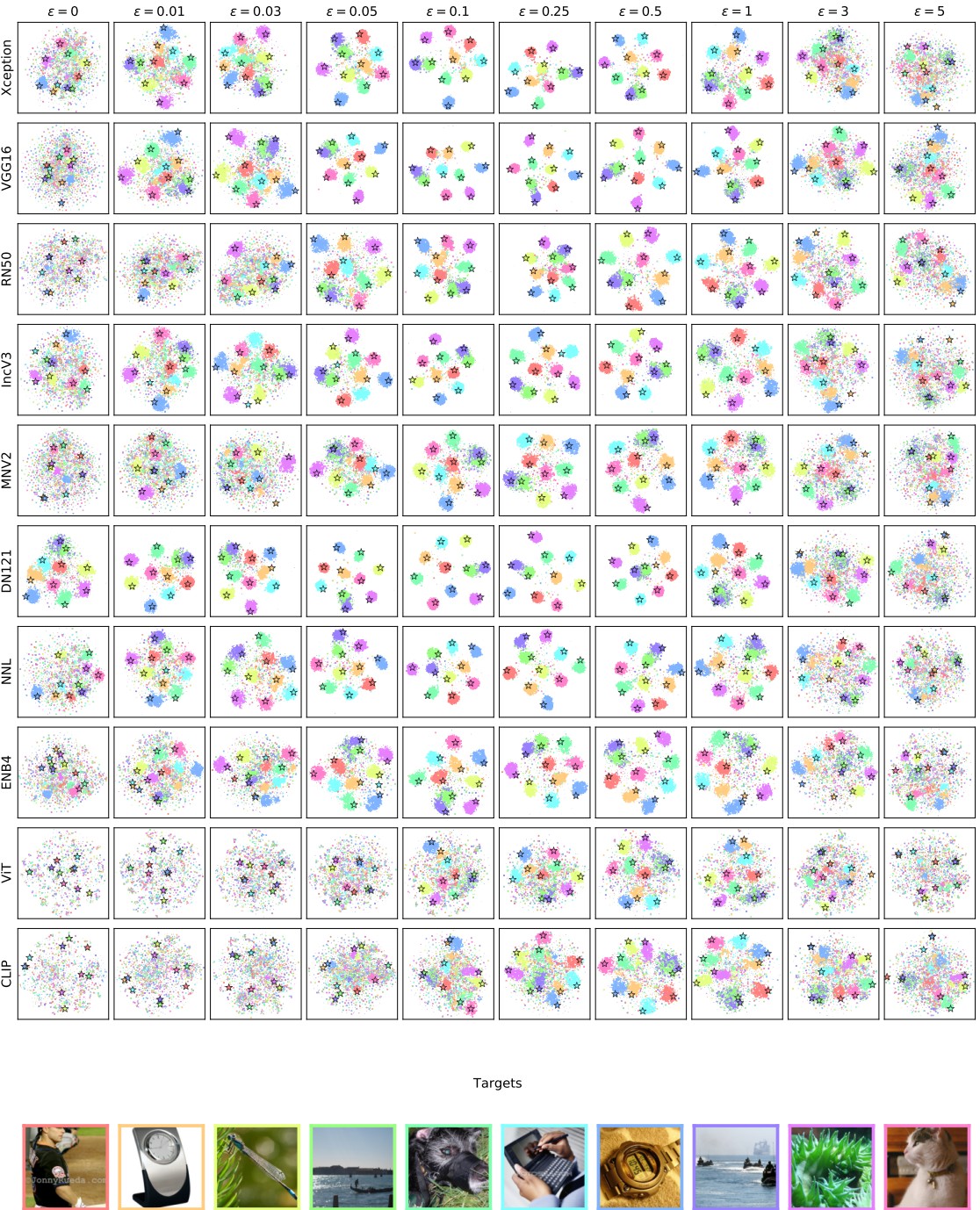

*Figure 6.* Extended data: t-SNE plots of destination-network representations of representation-targeted adversarial examples generated by using whitebox ResNet50 models of specified $\varepsilon$-robustness. (Best viewed in color and magnified.)

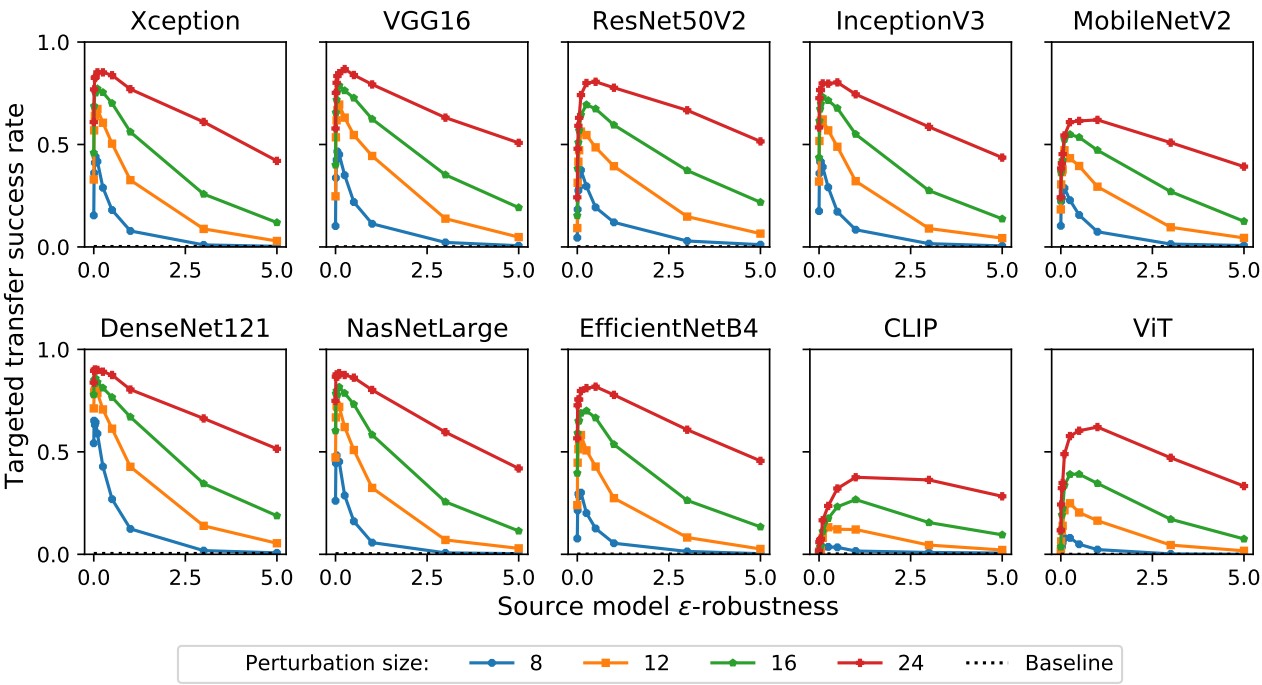

*Figure 7.* Extended ImageNet data (note extended horizontal axis in comparison with Figure 1): Transfer success rate of destination networks (CIFAR-10 classifiers) evaluated on targeted transferable adversarial examples using $\varepsilon$-robust ResNet50 source models with perturbation size $\|\delta\|_\infty \leq {}^{16}/{}_{256}$. Higher is a more successful attack. Baseline refers to the transfer rate of unperturbed images. (Best viewed in color.)

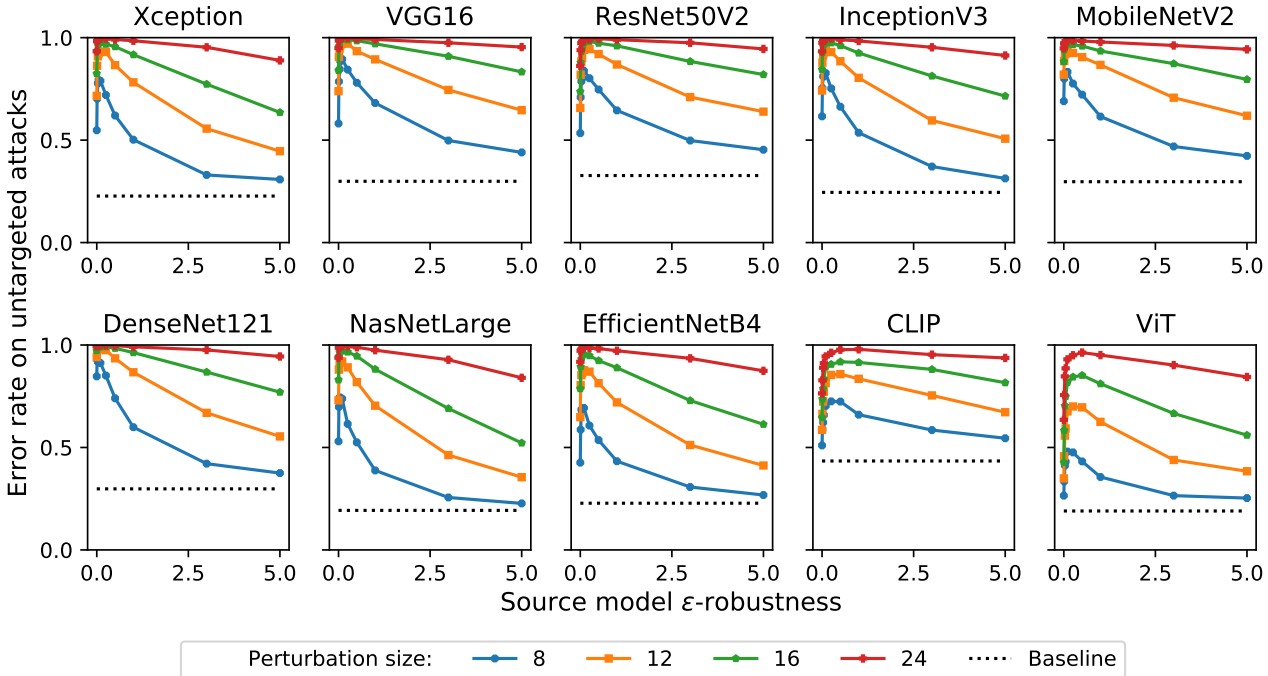

*Figure 8.* Extended ImageNet data (note extended horizontal axis in comparison with Figure 4). Error rate of destination networks (CIFAR-10 classifiers) evaluated on untargeted transferable adversarial examples using $\varepsilon$-robust ResNet50 source models with perturbation size $\|\delta\|_\infty \leq {}^{16}/{}_{256}$. Higher is a more successful attack. Baseline refers to the misclassification rate of unperturbed images. (Best viewed in color.)

## D. CIFAR-10 Data

We extend our experiments to the CIFAR-10 dataset to confirm that our results are general. We present the effectiveness of targeted and untargeted transferable adversarial examples (Figures 9 and 10). In addition, we present t-SNE plots of the destination-network representations of representation-targeted examples (Figure 11), as well as the cosine similarity between feature representations and the target images (Table 3). For all experiments, our results are not as exaggerated as with the ImageNet data, but nonetheless, we observe an increase in transferability of both class-targeted, untargeted, and representation-targeted adversarial examples when we use slightly-robust source networks, confirming that our claims generalize to networks trained on the CIFAR-10 dataset.

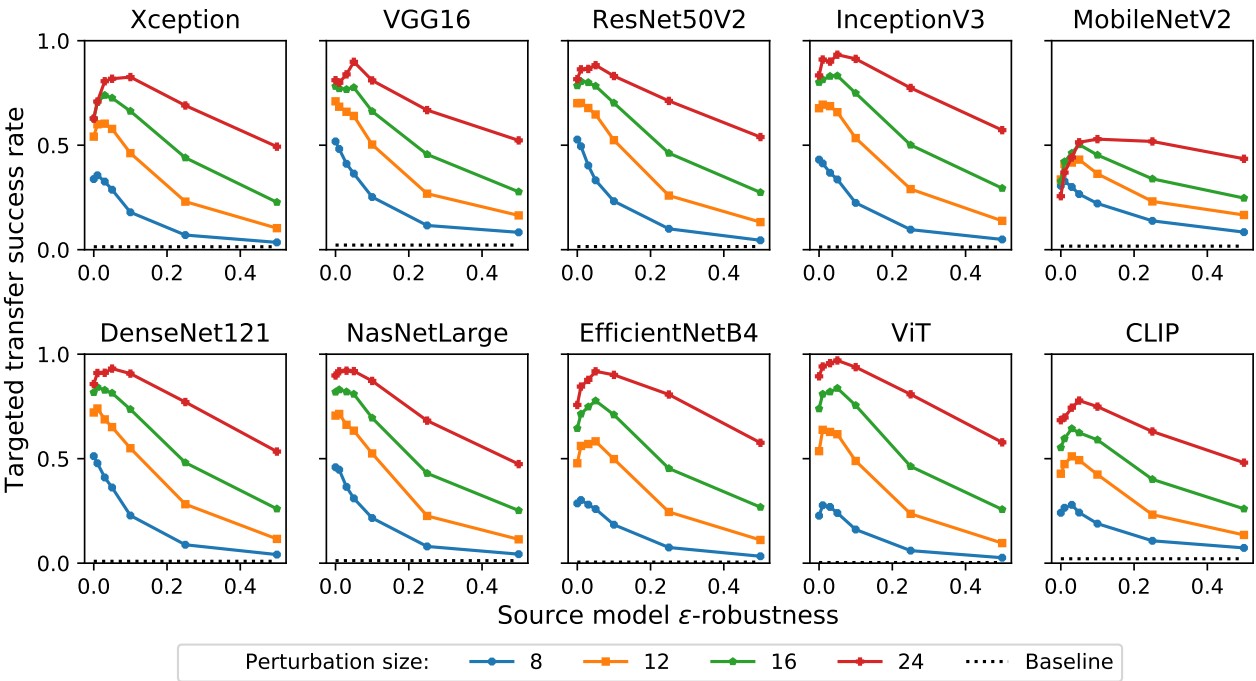

*Figure 9.* CIFAR-10 data: Transfer success rate of destination networks (CIFAR-10 classifiers) evaluated on targeted transferable adversarial examples using $\varepsilon$-robust ResNet50 source models with perturbation size $\|\delta\|_\infty \leq {}^{16}/_{256}$. Higher is a more successful attack. Baseline refers to the transfer rate of unperturbed images. (Best viewed in color.)

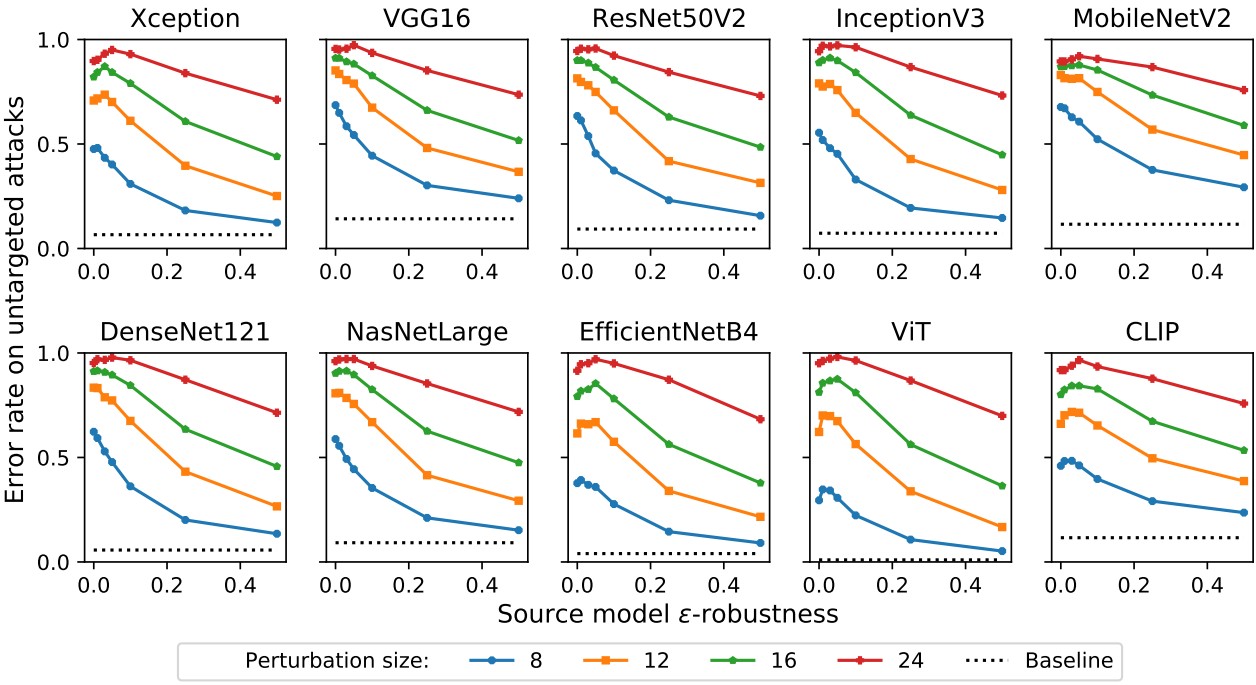

*Figure 10.* CIFAR-10 data: Error rate of destination networks (CIFAR-10 classifiers) evaluated on untargeted transferable adversarial examples using $\varepsilon$-robust ResNet50 source models with perturbation size $\|\delta\|_\infty \leq {}^{16}/{}_{256}$. Higher is a more successful attack. Baseline refers to the misclassification rate of unperturbed images. (Best viewed in color.)

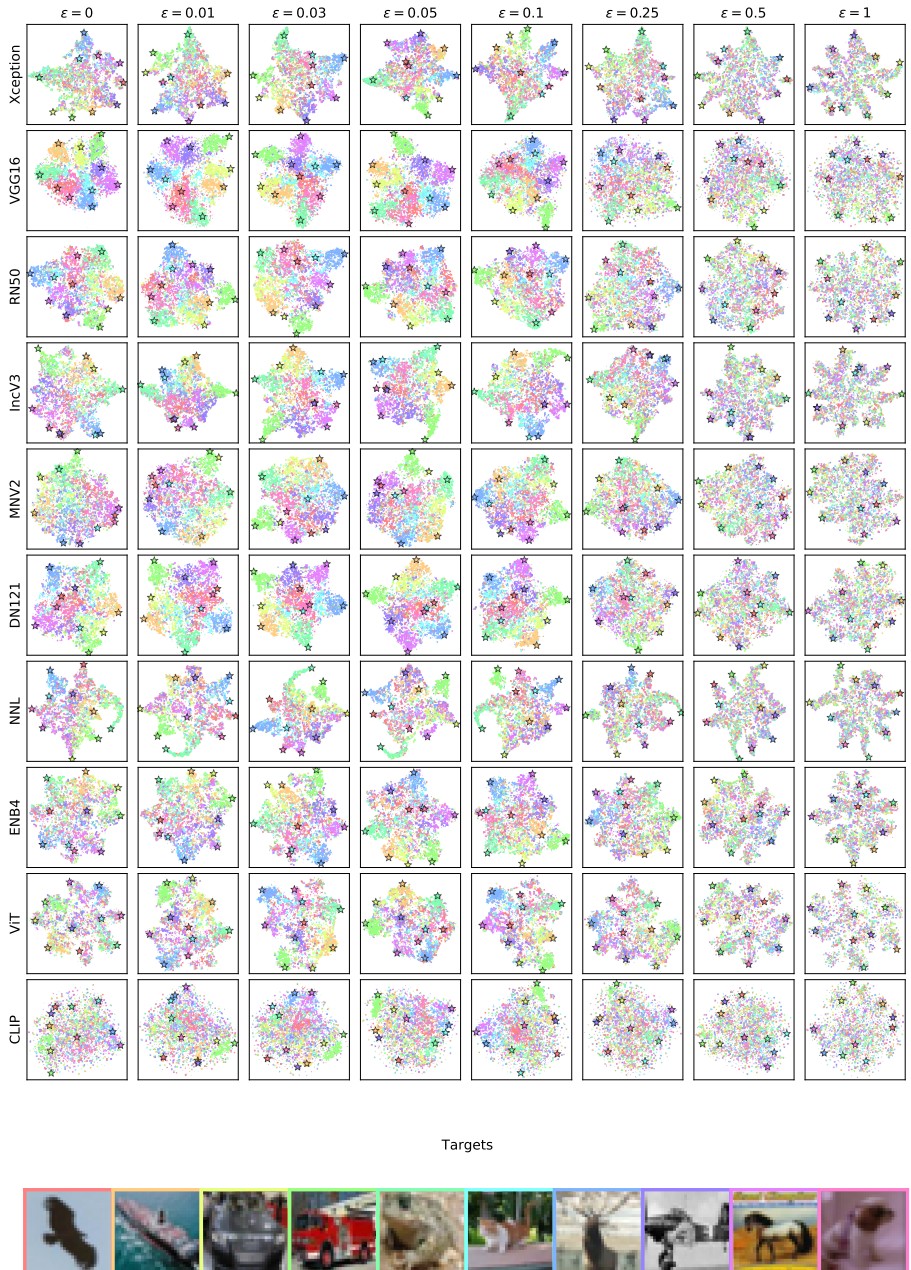

Targets

*Figure 11.* CIFAR-10 data: t-SNE plots of destination-network representations of representation-targeted adversarial examples generated by using whitebox ResNet50 models of specified $\varepsilon$-robustness. (Best viewed in color and magnified.)

*Table 3.* CIFAR-10 data: Cosine similarity between feature representations of representation-targeted adversarial examples and the targeted original images by robustness parameter of source model.

|  | 0 | 0.01 | 0.03 | 0.05 | 0.1 | 0.25 | 0.5 | 1 |
|---|---|---|---|---|---|---|---|---|
| Xception | 0.609 | 0.633 | 0.665 | **0.667** | 0.655 | 0.608 | 0.552 | 0.494 |
| VGG16 | 0.736 | 0.744 | **0.749** | 0.749 | 0.728 | 0.697 | 0.668 | 0.638 |
| RN50 | 0.691 | 0.709 | **0.717** | 0.715 | 0.705 | 0.652 | 0.600 | 0.546 |
| IncV3 | 0.662 | 0.686 | **0.706** | 0.700 | 0.695 | 0.637 | 0.590 | 0.532 |
| MNV2 | 0.630 | 0.647 | 0.664 | **0.670** | 0.667 | 0.644 | 0.615 | 0.563 |
| DN121 | 0.714 | 0.726 | **0.739** | 0.739 | 0.727 | 0.683 | 0.639 | 0.595 |
| NNL | 0.653 | 0.682 | **0.714** | 0.694 | 0.686 | 0.627 | 0.581 | 0.535 |
| ENB4 | 0.483 | 0.509 | 0.545 | **0.548** | 0.536 | 0.484 | 0.424 | 0.353 |
| ViT | 0.269 | 0.324 | **0.375** | 0.370 | 0.362 | 0.295 | 0.211 | 0.134 |
| CLIP | 0.768 | 0.771 | **0.773** | 0.772 | 0.772 | 0.767 | 0.762 | 0.755 |

# E. Examples of Adversarial Examples

We include class- and representation-targeted adversarial examples that have a perturbation generated with TMDI-FGSM and an $\ell_\infty$ constraint of $16/255$ (Figures 12 and 13).

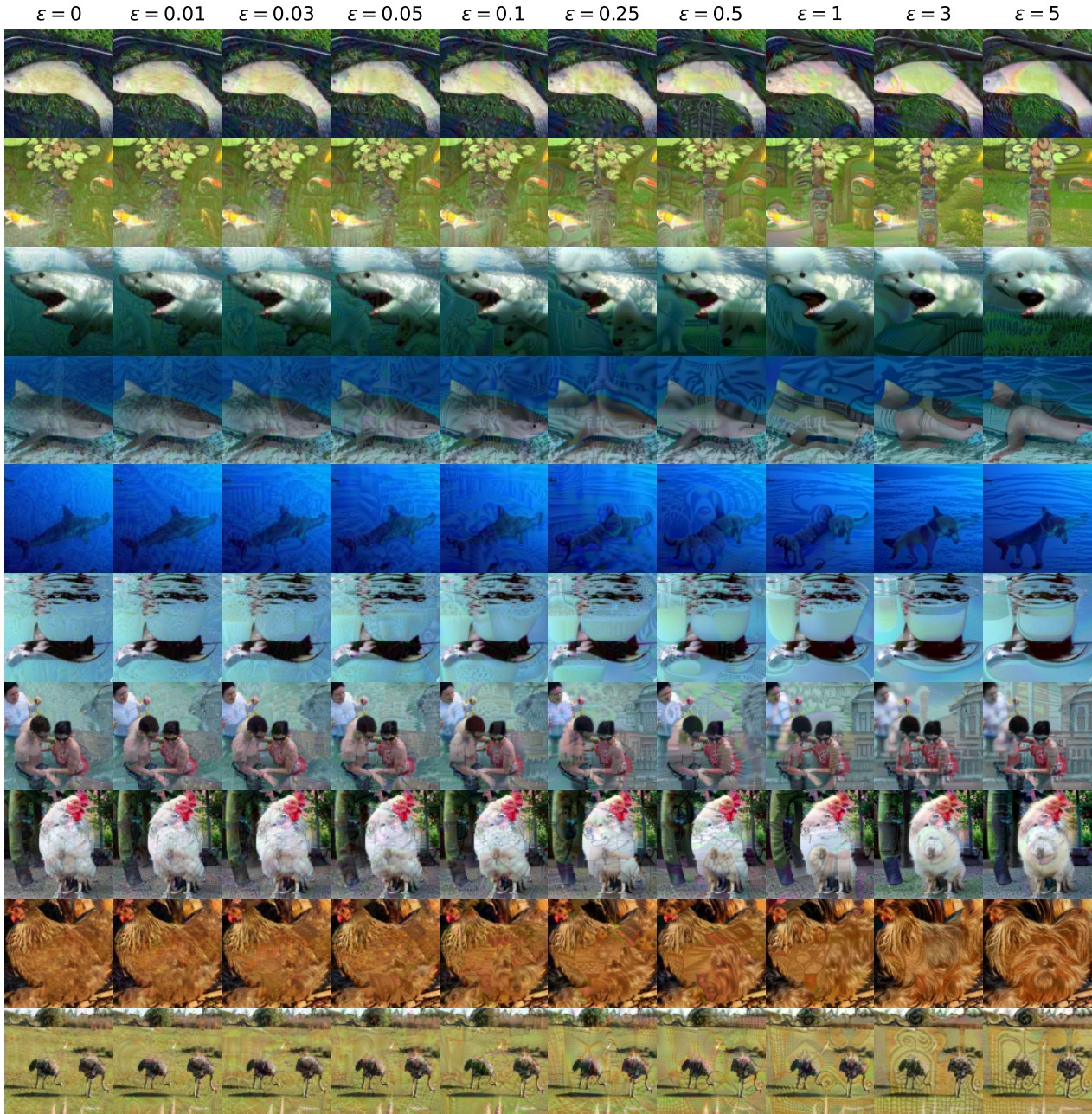

Figure 12. Examples of class-targeted adversarial examples, where the horizontal axis represents the robustness of the source network used to generate the adversarial examples. The adversarial perturbations are subject to an $\ell_\infty$ constraint of $16/255$, and are optimized with the TMDI-FGSM algorithm. (Best viewed in color.)

| $\varepsilon = 0$ | $\varepsilon = 0.01$ | $\varepsilon = 0.03$ | $\varepsilon = 0.05$ | $\varepsilon = 0.1$ | $\varepsilon = 0.25$ | $\varepsilon = 0.5$ | $\varepsilon = 1$ | $\varepsilon = 3$ | $\varepsilon = 5$ |

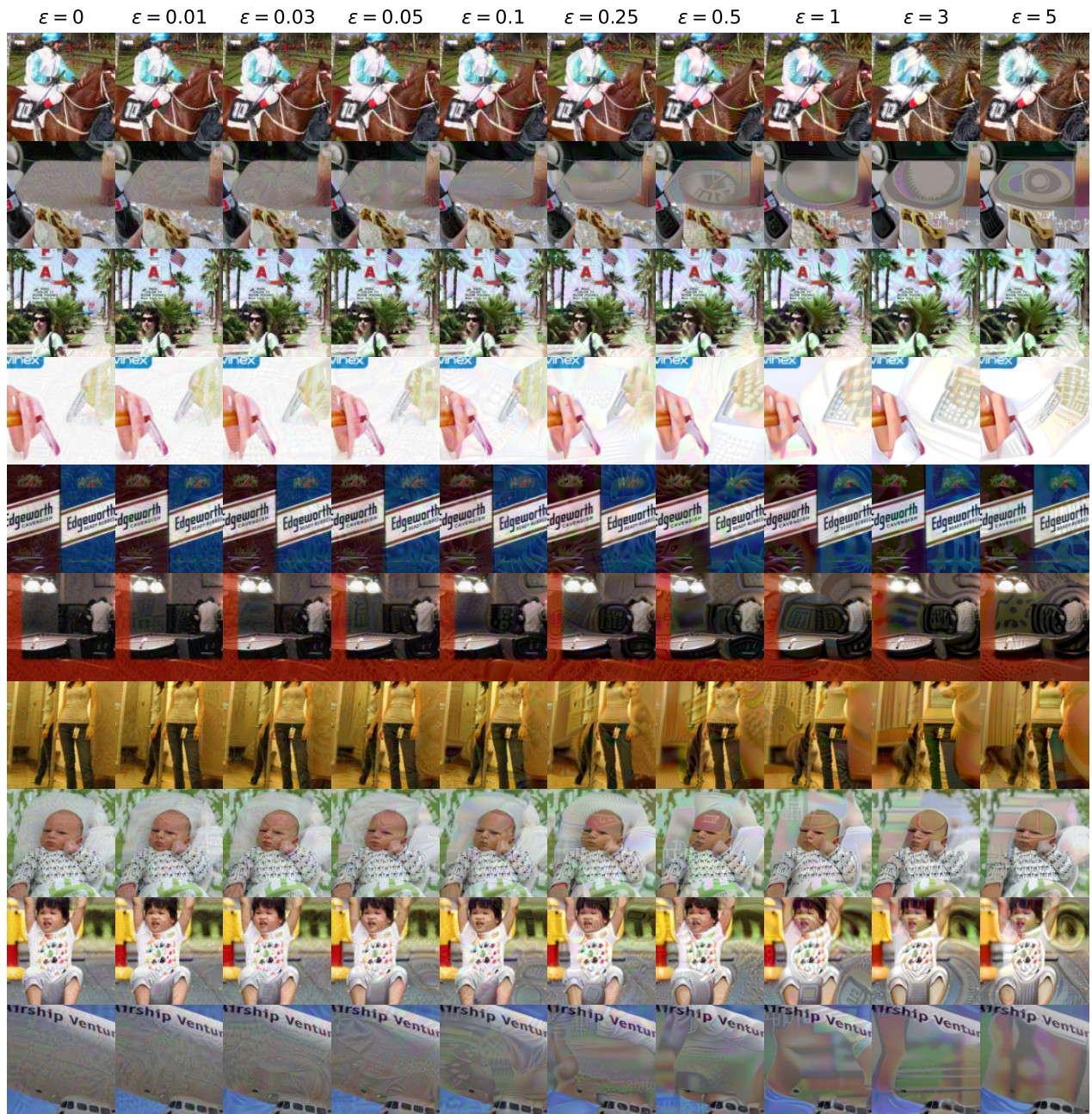

*Figure 13.* Examples of representation-targeted adversarial examples, where the horizontal axis represents the robustness of the source network used to generate the adversarial examples. The adversarial perturbations are subject to an $\ell_\infty$ constraint of $16/255$, and are optimized with the TMDI-FGSM algorithm. (Best viewed in color.)