# OpenReview forum: "Uncovering Universal Features: How Adversarial Training Improves Adversarial Transferability"
_ICML.cc/2021/Workshop/AML — ICML 2021 Workshop AML Poster_

### Official Review · Reviewer_sgoT · 2021-06-19
**The phenomenon is interesting. More theoretical analysis should be included.**

**Rating:** Accept
**Confidence:** 4

**Review:**

In this work, the authors show that adversarial examples can greatly transfer between slightly-robust networks and target networks. The paper show that these adversarial examples can transfer representation layer features substantially better than adversarial examples generated with non-robust networks.  The authors argue that this result supports a non-intuitive hypothesis that slightly robust networks exhibit universal features.

Pros: The uncovered phenomenon is really interesting. Experiment results are given to support the argument.

Cons: Although the phenomenon is really interesting, it is non-intuitive. More theoretical analysis should be included to demonstrate why this would happen.

---

### Decision · Program_Chairs · 2021-06-21

**Decision:**

Accept (Poster)

**Comment:**

This paper showed that can greatly transfer between slightly-robust networks and target networks. The finding is interesting and might have inspire future work.